

# Coherent deflection of atomic samples and positional mesoscopic superpositions

**L. F. Alves da Silva, L. M. R. Rocha and M. H. Y. Moussa**

Instituto de Física de São Carlos, Universidade de São Paulo,
P.O. Box 369, São Carlos, 13560-970, São Paulo, Brazil

## Abstract

We present a protocol based on the interplay between superradiance and superabsorption to achieve the coherent deflection of an atomic sample due to the momentum transfer from the atoms to a cavity field. The coherent character of this momentum transfer, causing the atomic sample to deflect as a whole, follows from the collective nature of the atomic superradiant pulse and its superabsorption by the cavity field. The protocol is then used for the construction of positional mesoscopic atomic superpositions in cavity quantum electrodynamics.

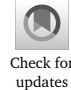

## 1 Introduction

The optical Stern-Gerlach effect — the splitting of the trajectory of an on- or off-resonant two-level atom by a quantized electromagnetic field —, dates to the late 1970s [1–3] and early

1980s [4], and its experimental demonstration occurred in the early 1990s [5]. Knowing that the photon statistics of a cavity field can manifest itself in the momentum distribution of the scattered atoms [6], the optical Stern-Gerlach was used for quantum nondemolition measurement of photon statistics [7] and for the state tomography of a cavity field [8,9]. Later, [10], the splitting of the atomic trajectory was considered for the proposition of a fully quantum protocol for two-dimensional atomic lithography, and also for entanglement detection from atomic deflection [11].

In the present work we propose a protocol for the coherent deflection of an atomic sample in cavity quantum electrodynamics, from which we can then construct, for example, a positional mesoscopic superposition. The generation of mesoscopic superpositions has been a much pursued challenge for their interface between the micro and macrophysics, allowing both the testing of fundamental quantum principles and applications in quantum technology [12–18].

To achieve such a coherent deflection of the sample, we take advantage of three previous developments: *i*) First, the optical Stern-Gerlach effect [1–5]. *ii*) Second, the recently proposed interplay between superradiance [19] and superabsorption [20] of a moderately dense atomic sample trapped inside a cavity [21]: When preparing the $N$-atoms sample in a superradiant state, with the cavity field in the vacuum, the coherent pulse emitted by the sample is superabsorbed by the resonant cavity mode due to an atom-field Rabi coupling $g$ enhanced by the factor $\sqrt{N}$. (This enhanced coupling was previously derived through a semiclassical approach [22, 23] and experimentally confirmed in what is called the ringing regime of superradiance [24].) The field excitation is then superradiated and superabsorbed back to the atomic sample in a cyclic decaying process. We demonstrate here that this superradiance-superabsorption interplay accounts for the momentum transfer between atoms and field, allowing the coherent deflection of the sample. *iii*) To know the states of the atomic sample and the cavity mode, which are necessary for computing the momentum transfer and for the construction of positional mesoscopic superpositions, we also resort to the Lewis & Riesenfeld dynamical invariants [25], as advanced in Ref. [26] for the atomic sample state, and in Ref. [27] for the field state.

## 2 The nonlinear mean-field Hamiltonians

As anticipated above, in Ref. [21] the authors consider a moderately dense atomic sample trapped inside a high-$Q$ cavity, resonantly interacting with a cavity mode. The Hamiltonian of the system, $H = H_0 + H_I$, is given by ($\hbar = 1$)

$$H_0 = \omega a^\dagger a + \omega S_z + \sum_k \omega_k b_k^\dagger b_k \,, \tag{1a}$$

$$H_I = g\left(S_+ a + S_- a^\dagger\right) + \sum_k \lambda_k \left(S_+ b_k + S_- b_k^\dagger\right) \,, \tag{1b}$$

with $H_0$ accounting for the cavity mode of frequency $\omega$ (described by the field creation and annihilation operators, $a^\dagger$ and $a$), resonant to the atomic sample (described by the collective pseudospin operator $S_z = \sum_{n=1}^N \sigma_z$). $H_0$ also accounts for the multimode reservoir of frequencies $\omega_k$ (described by the creation and annihilation operators, $b_k^\dagger$ and $b_k$). $H_I$ describes the interaction of the atomic sample, where $S_\pm = \sum_{n=1}^N \sigma_\pm$, with the cavity mode and the environment with coupling frequencies $g$ and $\lambda_k$, respectively. Considering the field quadratures $X_1 = \left(a + a^\dagger\right)/2$ and $X_2 = \left(a - a^\dagger\right)/2i$, a mean-field treatment of the system reduces the

Hamiltonian (1) to the nonlinear time-dependent form $H(t) = H_a(t) + H_f(t)$, with

$$H_a = \omega s_z + 2\Lambda_R \left( \langle s_x \rangle s_x + \langle s_y \rangle s_y \right) - 2\Lambda_I \left( \langle s_x \rangle s_y - \langle s_y \rangle s_x \right), \tag{2a}$$

$$H_f = \omega a^\dagger a + 2\sqrt{N} g \left( \langle s_x \rangle X_1 - \langle s_y \rangle X_2 \right). \tag{2b}$$

The atomic sample is thus replaced by a representative atom, described by $H_a$ through the operators $s_\mu = \sigma_\mu/2$, with $\mu = x, y, z$ and $\sigma_\pm = \left( \sigma_x \pm i\sigma_y \right)/2$. This atom is under a nonlinear amplification with strength $\Lambda = \Lambda_R + i\Lambda_I$, where

$$\Lambda_R = \sqrt{N} g \frac{\langle s_x \rangle \langle X_1 \rangle - \langle s_y \rangle \langle X_2 \rangle}{\langle s_x \rangle^2 + \langle s_y \rangle^2}, \tag{3a}$$

$$\Lambda_I = \sqrt{N} g \frac{\langle s_x \rangle \langle X_2 \rangle + \langle s_y \rangle \langle X_1 \rangle}{\langle s_x \rangle^2 + \langle s_y \rangle^2} - \frac{N\gamma}{2}, \tag{3b}$$

$\gamma$ being the atomic decay factor. Moreover, an enhanced effective coupling $\sqrt{N} g$ emerges between the representative atom and the cavity field as described by $H_f$. Basically, the mean-field approximation is used, as a method to approach the master equation of our many-body system, composed of the atomic sample and the field. This method consists in tracing out all the degrees of freedom of $N - 1$ atoms, leaving us with the reduced master equation for a single representative atom interacting with the cavity field as described by Hamiltonians (2a) and (2b).

Starting with the atomic sample in an inverted populated state and the cavity mode in the vacuum, the environment then triggers the superradiant pulse which is superabsorbed by the mode due to the enhanced coupling. Another important feature of the nonlinear mean-field Hamiltonians $H_a$ and $H_f$, is that although they commute with each other, leading to separate Schrödinger equations, $i\partial_t |\psi_\xi\rangle = H_\xi |\psi_\xi\rangle$ ($\xi = a$ or $f$), for initial product states $|\psi_a\rangle \otimes |\psi_f\rangle$, there is an indirect coupling between atom and field coming from the time-dependent mean values.

## 3 The Lewis & Riesenfeld dynamic invariants

To solve the Schrödinger equation for the time-dependent Hamiltonians $H_a$ and $H_f$, we use the Lewis & Riesenfeld dynamic invariants [25], for the atom $I_a(t)$ and the field $I_f(t)$, defined as $\partial_t I_\xi - i \left[ I_\xi, H_\xi \right] = 0$. Following Refs. [26, 27], we propose the operators

$$I_a = \langle s_x \rangle s_x + \langle s_y \rangle s_y + \langle s_z \rangle s_z, \tag{4a}$$

$$I_f = a^\dagger a - 2\langle X_1 \rangle X_1 - 2\langle X_2 \rangle X_2 + \chi, \tag{4b}$$

to obtain the system

$$\langle \dot{s}_x \rangle = -\omega \langle s_y \rangle + 2\langle s_z \rangle \left( \Lambda_R \langle s_y \rangle - \Lambda_I \langle s_x \rangle \right), \tag{5a}$$

$$\langle \dot{s}_y \rangle = \omega \langle s_x \rangle - 2\langle s_z \rangle \left( \Lambda_R \langle s_x \rangle + \Lambda_I \langle s_y \rangle \right), \tag{5b}$$

$$\langle \dot{s}_z \rangle = \Lambda_I \left( \langle s_x \rangle^2 + \langle s_y \rangle^2 \right), \tag{5c}$$

$$\langle \dot{X}_1 \rangle = \omega \langle X_2 \rangle - \sqrt{N} g \langle s_y \rangle, \tag{5d}$$

$$\langle \dot{X}_2 \rangle = -\omega \langle X_1 \rangle - \sqrt{N} g \langle s_x \rangle, \tag{5e}$$

together with the equation $\dot{\chi} = -\sqrt{N} g \left( \langle s_x \rangle \langle X_2 \rangle + \langle s_y \rangle \langle X_1 \rangle \right)$ that avoids unnecessary constraints on the mean values defining $I_f$.

From the fact that $\langle \dot{I}_a \rangle = 0$, such that $\langle I_a \rangle = \langle s_x \rangle^2 + \langle s_y \rangle^2 + \langle s_z \rangle^2 = R^2$, we consider a Bloch sphere of radius $R$ to define the mean values $\langle s_x \rangle = R \sin\theta \cos\phi$, $\langle s_y \rangle = R \sin\theta \sin\phi$, $\langle s_z \rangle = R \cos\theta$. We then derive the eigenvectors of $I_a$, given by $|+,t\rangle = \cos(\theta/2)|e\rangle + e^{i\phi}\sin(\theta/2)|g\rangle$ and $|-,t\rangle = \sin(\theta/2)|e\rangle - e^{i\phi}\cos(\theta/2)|g\rangle$, where the vector $|g\rangle$ ($|e\rangle$) of the representative atom corresponds to the entire sample in the ground (excited) state. Starting from the general superposition $|\psi_a(t)\rangle = c_+ e^{i\Phi_+^a(t)}|+,t\rangle + c_- e^{i\Phi_-^a(t)}|-,t\rangle$, where the Lewis & Riesenfeld phase factors are given by

$$\Phi_+^a(t) = -\frac{\omega t}{2} - \int_0^t \Lambda_R \sin^2(\theta/2)\, dt', \tag{6a}$$

$$\Phi_-^a(t) = -\frac{\omega t}{2} + \int_0^t \Lambda_R \cos^2(\theta/2)\, dt', \tag{6b}$$

we verify that the eigenstate $|-,t\rangle$ is ruled out of the solution of the atomic Schrödinger equation by the self-consistency condition

$$\langle s_z \rangle = \left[ \left(|c_+|^2 - |c_-|^2\right)\cos\theta + 2\,\mathrm{Re}\left(c_+ c_-^* e^{i(\Phi_+^a - \Phi_-^a)}\right)\sin\theta \right]/2 = R\cos\theta\,.$$

For a positive defined radius $R = 1/2$, this condition leads to $|c_+| = 1$ and $|c_-| = 0$, in agreement with the self-consistency conditions for $\langle s_x \rangle$ and $\langle s_y \rangle$. We then derive the solution of the Schrödinger equation for the atom as

$$|\psi_a(t)\rangle = e^{i\Phi_+^a(t)}|+,t\rangle, \tag{7}$$

while the solution for the field is reached by defining $\alpha = \langle a \rangle$ [27], with

$$\dot{\alpha} = -i\left[\omega\alpha + \left(\sqrt{N}g/2\right)\sin\theta\, e^{-i\phi}\right],$$

in agreement with Eqs. (5d) and (5e). Starting with the field in the coherent state $\alpha_0$, we then obtain

$$|\psi_f(t)\rangle = e^{i\Phi^f(t)}|\alpha(t)\rangle, \tag{8}$$

with the Lewis & Riesenfeld phase

$$\Phi^f(t) = \int_0^t \left[\omega|\alpha|^2 - \frac{\sqrt{N}g}{4}\left(\alpha e^{i\phi} + \alpha^* e^{-i\phi}\right)\sin\theta\right] dt'. \tag{9}$$

After computing the system state vector

$$|\psi(t)\rangle = e^{i\left[\Phi_+^a(t) + \Phi^f(t)\right]}|+,t\rangle \otimes |\alpha(t)\rangle, \tag{10}$$

we are now able to approach the coherent deflection of the atomic sample, starting with some considerations on the experimental implementation of the process, schematically illustrated in Fig. 1. As shown in Fig. 1(a), we must assume that the trapped sample, with the atoms initially in their ground states, is placed on a node of the standing-wave field as in [8], for a greater atomic momentum transfer, proportional to the gradient of the cavity field [28]. Then, by manipulating the convexity of the trap potential, as drawn in Fig. 1(b), a moderately dense atomic sample is built. The initial state of the sample is then immediately prepared in the superposition $|\psi_a(0)\rangle = \cos[\theta_0/2]|e\rangle + e^{i\phi_0}\sin[\theta_0/2]|g\rangle$ [29]. Right after the preparation of the state $|\psi_a(0)\rangle$, the trap potential is turned off and the sample starts to interact with the cavity undergoing superradiance as illustrated in Fig. 1(c). Finally, as in Fig. 1(d), the sample leaves the cavity under gravity or, alternatively, we can consider a sample of ions accelerated by an electric field which is turned on immediately after the radiation-matter interaction. Starting from the cavity mode in the vacuum, we thus have the initial state $|\psi(0)\rangle = |+,0\rangle \otimes |0\rangle$, which evolves exactly to $|\psi(t)\rangle$ in Eq. (10).

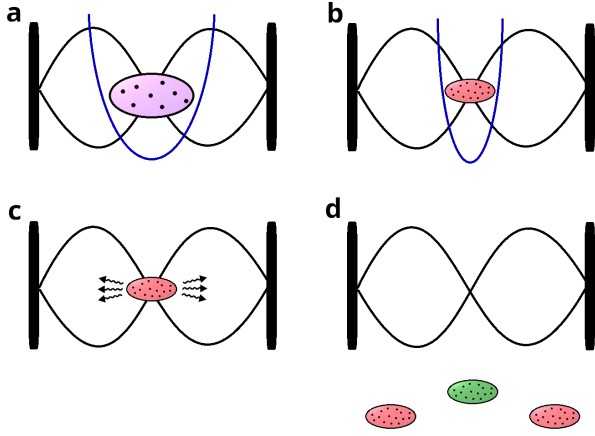

Figure 1: Schematic illustration of the experimental realization of the coherent deflection of an atomic sample initially trapped inside a cavity. In ($a$), the trapped atoms in their ground states are placed on a node of the cavity mode, while in ($b$) a moderately dense atomic sample is built, controlling the convexity of the trap potential, and the initial state of the sample is prepared. In ($c$), the trap potential is turned off and the sample starts to interact with the cavity undergoing superradiance. In ($d$), the sample leaves the cavity accelerated by gravity or an electric field, being deflected due to the coherent momentum transfer to the cavity field.

## 4 Regimes of the superradiance-superabsorption interplay

For computing $\theta(t)$, $\phi(t)$ and $\alpha(t) = |\alpha(t)|e^{i\phi_\alpha(t)}$ from Eqs. (5), we consider the condition $\omega_0 \gg N\gamma, \sqrt{N}g$. We then derive the solution $\phi(t) \approx \pi/2 - \phi_\alpha(t) \approx \phi_0 + \omega t$, where $\tan\phi_\alpha = \langle X_2 \rangle / \langle X_1 \rangle \approx \tan(\pi/2 - \phi)$, and the Lienard system

$$\frac{d\theta}{dt} = \frac{N\gamma}{2}\sin\theta - 2\sqrt{N}g|\alpha|\,, \tag{11a}$$

$$\frac{d|\alpha|}{dt} = -\frac{\sqrt{N}g}{2}\sin\theta\,, \tag{11b}$$

leading to the Lienard equation

$$\ddot{\theta} = \frac{N\gamma}{2}\cos(\theta)\dot{\theta} + (\sqrt{N}g)^2\sin\theta\,, \tag{12}$$

which helps us to define, regarding the coherence parameter $\epsilon = 4\sqrt{N}g/N\gamma$, three regimes for solutions of our superradiance-superabsorption interplay: the overdamped ($\epsilon \ll 1$), the damped ($\epsilon \approx 1$), and the underdamped ($\epsilon \gg 1$) regimes. The definition of these regimes becomes clear by noting that the parameter $\epsilon$ follows from the competition between the effective oscillation frequency $\sqrt{N}g$ and the effective damping factor $N\gamma/4$, as shown in Eq. (12). As expected, greater coherence of the sample deflection results from greater atom-field couplings and the smaller samples and atomic decay factors.

We first consider the overdamped regime where an approximated analytic solution can be obtained from Eqs. (11), which also applies, with much less accuracy, for the damped regime. In this regime the superradiant-superabsorption cycle begin to emerge, indicating that the excitation superradiated by the sample is superabsorbed by the cavity field, ensuring the momentum transfer between radiation and matter. We then consider the underdamped regime where the momentum transfer is fully accomplished.

*i*) *The overdamped regime*. For the overdamped regime, the perturbative parameter $\epsilon$ allows us to consider the first order expansions $\theta(t) \approx \theta_h(t) + \epsilon \vartheta(t)$ and $|\alpha(t)| \approx \alpha_h(t) + \epsilon \tilde{\alpha}(t)$. The solutions $\sin \theta_h(t) = \text{sech}\left[(t - \tau_D)/\tau\right]$ [26] and $\alpha_h(t) = \alpha_0$, arise from the homogeneous equations resulting when we turn off the atom-field coupling, such that $\epsilon = 0$. To be computed below, $\tau_D$ is the delay time for the initial atomic state $|\psi_a(0)\rangle$ to evolve to the well-known superradiant superposition $\left(|e\rangle + e^{i\phi}|g\rangle\right)/\sqrt{2}$ [24], whereas $\tau = 2/N\gamma$ is the characteristic emission time of the free-sample Dicke's superradiance. These approximations reduce the Lienard system to the decoupled equations $\dot{\vartheta} = (N\gamma/2)(\vartheta \cos \theta_h - 4\alpha_h)$ and $\dot{\alpha} = -(N\gamma/2)\sin \theta_h$, leading to the solutions

$$\theta(t) \approx \theta_h(t) + \alpha_0 \epsilon \cos \theta_h(t), \tag{13a}$$

$$|\alpha(t)| \approx \alpha_0 + (\epsilon/4)\left[\theta_h(t) - \theta_0\right], \tag{13b}$$

where we have assumed the consistent boundary conditions $\theta(\tau_D) = \theta_h(\tau_D) = \pi/2$ and $\alpha(0) = \alpha_h(0) = \alpha_0$, such that $\vartheta(\tau_D) = 0$ and $\tilde{\alpha}(0) = 0$. We have also assumed $\phi_0 = \pi/2$ leading to $\phi_\alpha(0) = 0$ and $\alpha(0) = |\alpha(0)| = \alpha_0$. From Eq. (13a) we derive the expression

$$e^{\frac{\tau_D}{\tau}} \tan \frac{\theta_0}{2} + \alpha_0 \epsilon \left(\tan \frac{\theta_0}{2} - \sinh \frac{\tau_D}{\tau}\right) \tanh \frac{\tau_D}{\tau} \approx 1, \tag{14}$$

which enables us to compute the delay time $\tau_D$. Starting from the superradiant state with $\theta_0 = \pi/2$, we verify, as expected, that $\tau_D = 0$. For $\theta_0 \ll 1$ we may assume that $\tau_D/\tau \gg 1$, leading us from Eq. (14) to

$$\tau_D \approx \tau \ln \left| \frac{\cot(\theta_0/2) - \alpha_0 \epsilon}{1 - \alpha_0(\epsilon/2)\cot(\theta_0/2)} \right|, \tag{15}$$

showing that for $\epsilon = 0$ we retrieve the well-known result for the Dicke's superradiance: $\tau_D \approx \tau \ln \cot(\theta_0/2)$.

*ii*) *The underdamped regime*. Back to the Lienard equation (12), we now linearize the sinusoidal functions around $\theta = \pi$, to retrieve the standard solution for an underdamped oscillator, given by

$$\theta(t) \approx \pi - \left[(\pi - \theta_0)\cos\left(\sqrt{N}gt\right) + 2\alpha_0 \sin\left(\sqrt{N}gt\right)\right] e^{-N\gamma t/4}, \tag{16}$$

with $|\alpha(t)|$ following by substituting Eq. (16) into Eq. (11b). We note that the solutions in Eq. (13) could also have been derived from the linearization procedure around $\theta = \pi$ and $\alpha = 0$, but under the restriction that the initial condition is far from the metastable point $\theta_0 = 0$.

# 5 Coherent deflection of the sample and positional superpositions

With the above solutions in Eqs. (13) and (16), we analyze the evolution of the system state vector in Eq. (10), where we now consider the position dependence of the atom-field coupling $g(x) = \mu\mathcal{E}\sin(kx)$, with $\mu$, $\mathcal{E}$ and $k$ standing respectively for the atomic dipole moment, the effective electric field per photon, and the wave-vector of the cavity mode. As already anticipated, we assume that the trapped sample is placed on a node of the cavity field, such that $g(x) \approx \mu\mathcal{E}kx$, remembering that the superradiant sample must be small compared to the wavelength of the superabsorptive mode.

Starting from the initial state vector of the system

$$|\psi(x, t = 0)\rangle = \int_{-\infty}^{+\infty} \Theta(x)|+, 0\rangle \otimes |\alpha_0\rangle \otimes |x\rangle dx, \tag{17}$$

with $\Theta(x)$ standing for the spatial distribution of the atomic sample, we obtain after the interaction time $t$,

$$|\psi(x,t)\rangle = \int_{-\infty}^{+\infty} e^{+i[\Phi_+^a(x,t)+\Phi^f(x,t)]}\Theta(x)|+,t\rangle \otimes |\alpha(x,t)\rangle \otimes |x\rangle dx, \qquad (18)$$

now considering that the field coherent state also depends upon the Rabi frequency $g(x)$. The Raman-Nath regime — by which the kinetic energy of the sample is neglected, by assuming that its transverse displacement along the interaction time is small compared to the wavelength of the mode — is here perfectly observed since the sample is released from the trap with zero velocity. From the state vector in Eq. (18), we next analyse the momentum transfer for the overdamped and the underdamped regimes, considering the solutions $\Phi_+^a(t) = -\omega t/2$ and $\Phi^f(t) = 0$ valid whatever the regime.

*i) The overdamped regime.* By projecting the state $|\psi(x,t)\rangle$ onto the position space, we obtain the solution

$$|\psi(x,t)\rangle = \frac{e^{-i\omega t/2}}{\sqrt{2}}\Theta(x)\left[e^{i\theta_h(t)/2}e^{ikx\kappa(t)}|+,t\rangle + e^{-i\theta_h(t)/2}e^{-ikx\kappa(t)}|-,t\rangle\right] \otimes |\alpha(x,t)\rangle, \quad (19)$$

where we have defined the effective interaction parameter

$$\kappa(t) = \frac{2\mu\mathcal{E}\alpha_0}{\sqrt{N}\gamma}\tanh\left(\frac{t-\tau_D}{\tau}\right), \qquad (20)$$

and considered the expansion $|+,t\rangle = \left[e^{i\theta(x,t)/2}|+,t\rangle + e^{-i\theta(x,t)/2}|-,t\rangle\right]/\sqrt{2}$, with $|\pm,t\rangle = \left(|e\rangle \pm e^{i\phi(t)}|g\rangle\right)/\sqrt{2}$. The sample-field momentum transfer $k\kappa(t)$ may be alternatively computed from $\Delta\dot{p} = \sqrt{N}\vec{\mu}.\nabla\vec{E}$, where $\vec{E} = \mathcal{E}\alpha(t)\sin(kx)\hat{u}$ and $\sqrt{N}\vec{\mu}$ is an effective dipole moment. Then, it follows the rate $\kappa = \Delta p/k = \sqrt{N}\mu\mathcal{E}\alpha_0\Delta t$ which, for time intervals around the characteristic emission time $\tau = 2/N\gamma$, leads to $\kappa = \Delta p/k = 2\mu\mathcal{E}\alpha_0/\sqrt{N}\gamma$, in agreement with Eq. (20).

It is well-known in Dicke's superradiance that $\theta_0 = 0$ implies a metastable state of the atomic sample, of infinitely long duration. Here, as we conclude from the Lienard Eqs. (11), a metastable state of the radiation-matter system occurs for $\alpha_0 = 0$ and $\theta_0 = 0$. Regarding $\alpha_0$, we emphasize that our experiment does not require a high finesse cavity as far as the necessary superradiant-superabsorption cycle occurs in a short time interval of the order of $\tau_D + \tau \ll 1/\gamma$. However, the cavity must be cooled so that the initial average excitation of the field, $\alpha_0$, is small enough to ensure $\alpha_0\epsilon \ll 1$. Regarding the atomic variable $\theta_0$, we may consider, as an approximation, the result $\tau_D \approx \tau\ln[\cot(\theta_0/2)] \approx \tau\ln N$ from Dicke's superradiance [19], to infer that $\theta_0 \approx 2/N$.

For a time interval around $\tau_D + \tau$, such that $\tanh[(t-\tau_D)/\tau] \approx 1$ and $\kappa(t) \approx 2\mu\mathcal{E}\alpha_0/\sqrt{N}\gamma$, it is reasonable to disregard the dependence on position of the field state $\alpha(x,t)$, once the cavity field superabsorption has already been established. After Fourier transforming the state vector $|\psi(x,t)\rangle$ over the momentum representation, it follows that

$$|\psi(p,t)\rangle = \frac{1}{\sqrt{2}}\left[e^{-i\phi_+(t)}\mathcal{F}[p-k\kappa(t)]|+,t\rangle + e^{-i\phi_-(t)}\mathcal{F}[p+k\kappa(t)]|-,t\rangle\right] \otimes |\alpha(t)\rangle, \quad (21)$$

with $\phi_\pm(t) = [\omega t \pm \theta_h(t)]/2$ and the amplitude

$$\mathcal{F}(p) = \frac{1}{\sqrt{2\pi}}\int_{-\infty}^{+\infty} e^{-ipx}\Theta(x)\,dx, \qquad (22)$$

is the Fourier transform of the atomic spatial distribution. Eq. (21) shows that the sample is coherently deflected with momentum $\pm k\kappa(t)$ in the states $|\pm,t\rangle$.

*ii*) *The underdamped regime*. By inserting the solution 16 into Eq. (18) projected onto the position space, we obtain the Fourier transform

$$|\psi(p,t)\rangle = \left[e^{-i\omega t/2}\mathcal{F}_-(p)|e\rangle + ie^{i\omega t/2}\mathcal{F}_+(p)|g\rangle\right] \otimes |\alpha(t)\rangle, \tag{23}$$

where, using $R(t) \approx \sqrt{(\theta_0 - \pi)^2/4 + \alpha_0^2}e^{-N\gamma t/4}$ and $\tan\varphi = 4\alpha_0/\pi$, we obtain

$$\mathcal{F}_\pm(p) = \frac{i}{\sqrt{8\pi}} \int_{-\infty}^{+\infty} e^{-ipx}\Theta(x)\left(e^{-iR(t)\cos\left(\sqrt{N}\mu\mathcal{E}kxt+\varphi\right)} \pm e^{iR(t)\cos\left(\sqrt{N}\mu\mathcal{E}kxt+\varphi\right)}\right)dx. \tag{24}$$

From the Bessel identity [30]

$$e^{\pm iR\cos\zeta} = \sum_{n=-\infty}^{\infty} (\pm i)^n J_n(R)e^{\mp in\zeta}, \tag{25}$$

and considering $R(t) \ll 1$, in accordance with the linearization procedure, we finally obtain

$$\mathcal{F}_+(p) \approx iJ_0(R)\mathcal{F}(p), \tag{26a}$$

$$\mathcal{F}_-(p) \approx J_{+1}(R)\left[e^{i\varphi}\mathcal{F}\left(p - \sqrt{N}\mu\mathcal{E}kt\right) + e^{-i\varphi}\mathcal{F}\left(p + \sqrt{N}\mu\mathcal{E}kt\right)\right], \tag{26b}$$

with $J_0(R) = 1 - (R/2)^2$, $J_{+1}(R) = -J_{-1}(R) \approx R/2$. From Eqs. (26) we verify the splitting of the whole incident sample into three different paths. The undeflected path is associated with the representative state $|g\rangle$ whereas the deflected ones, with momenta $\pm\sqrt{N}\mu\mathcal{E}kt$, are associated with $|e\rangle$:

$$|\psi(p,t)\rangle \approx \left\{i\mathcal{F}(p)|g\rangle + (R/2)\left[e^{i\varphi}\mathcal{F}\left(p - \sqrt{N}\mu\mathcal{E}kt\right) + e^{-i\varphi}\mathcal{F}\left(p + \sqrt{N}\mu\mathcal{E}kt\right)\right]|e\rangle\right\} \otimes |\alpha(t)\rangle. \tag{27}$$

We observe that the momentum transfer is that of a single atom [8,9] multiplied by the factor $\sqrt{N}$, which is larger the longer the sample-field interaction time, i.e., the larger the number of superradiance-superabsorptive cycles. If on the one hand the momentum transfer increases with time, on the other the measurement probability of the deflected sample decreases as a function of the damping function $R(t)$.

## 6 Numerical analysis for cavity-QED experimental scenario

Next, for both the overdamped and the underdamped regimes, we must characterize the superradiant-superabsorption cycles and estimate the magnitude of the sample-field momentum transfer. Considering the energies of the representative atom and the cavity field, given by $\varepsilon_a = \omega_0 \langle\sigma_z\rangle/2$ and $\varepsilon_f = \omega_0 \langle a^\dagger a\rangle$, we compute the complementary intensities $\mathcal{I}_a$ and $\mathcal{I}_f$ [21]:

$$\mathcal{I}_a = -N\frac{d\varepsilon_A}{dt} = N\omega_0|\langle\sigma_-\rangle|\left[N\gamma|\langle\sigma_-\rangle| + 2\sqrt{N}g\sin(\phi_\sigma - \phi_a)|\langle a\rangle|\right], \tag{28a}$$

$$\mathcal{I}_f = -N\frac{d\varepsilon_F}{dt} = N^2\gamma\omega_0|\langle\sigma_-\rangle|^2 - \mathcal{I}_A. \tag{28b}$$

For the implementation of the overdamped, damped, and underdamped regimes, we consider $\omega = 10^5 g$ and $\gamma = 5 \times 10^{-3} g$, in order to contemplate both microwave [17] and optical [31] cavity-QED regimes, simultaneously. In fact, depending on the value of the Rabi frequency, $g \approx 10^5 Hz$ or $\approx 10^9 Hz$, we are in the microwave or optical regime, respectively. Starting with the overdamped regime, considering a sample with $N = 10^8$ atoms, such that $\epsilon \approx 0.1$, in Fig. 2(a, b, and c) we set $\alpha_0 = 0.1$, $\theta_0 = 2/N$, and $\phi_0 = \pi/2$, to draw the curves for

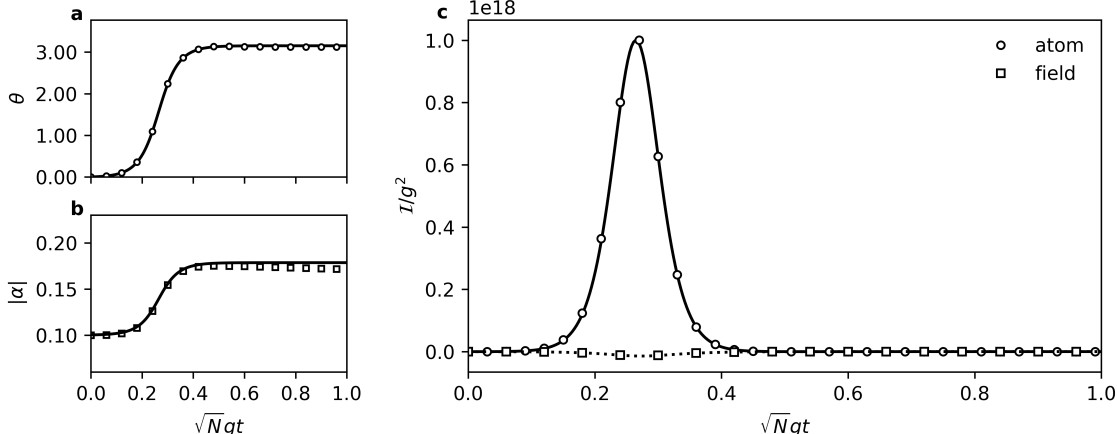

Figure 2: Plot of the numerical and analytical solutions for (a) $\theta(t)$, (b) $|\alpha(t)|$, (c) $\mathcal{I}_a$ and $\mathcal{I}_f$, against $\sqrt{N}gt$, for the overdamped regime: $N = 10^8$ and $\epsilon \approx 0.1$. We have assumed $\omega = 10^5 g$, $\gamma = 5 \times 10^{-3} g$, $\alpha_0 = 0.1$, $\theta_0 = 2/N$ and $\phi_0 = \pi/2$. The circles and squares represent the numerical solutions whereas the full and dotted lines represent the analytical ones.

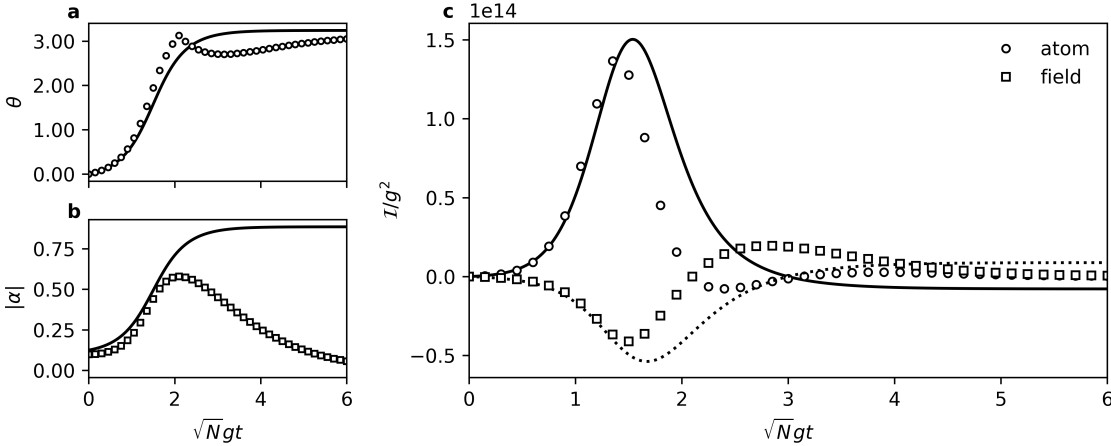

Figure 3: The same as in Fig. 2 for the damped regime: $N = 10^6$ and $\epsilon \approx 1$.

the numerical and analytical solutions for $\theta(t)$, $|\alpha(t)|$, $\mathcal{I}_a$, and $\mathcal{I}_f$, respectively. Whereas the circles and squares represent the numerical solutions, the full and dotted lines represent the analytical ones. As we observe, the analytical solutions match very well for the overdamped regime where we basically observe, in Fig. 2(c), an atomic superradiant pulse with intensity of about $10^{18} g^2$ and delay time $\tau_D \approx 2.65 \times 10^{-4} g^{-1}$, in perfect agreement with the analytical value coming from Eq. (15). The field superabsorption, presenting negative intensity [21], is inhibited by the small coherence parameter $\epsilon$. In Fig. 3($a$, $b$, and $c$), we plot the same functions as in Fig. 2 for the damped regime, with $N = 10^6$ such that $\epsilon \approx 1$, and all other parameters equal to those in Fig. 2. As anticipated above, our overdamped solutions apply with much less accuracy to the damped regime. We now observe a superradiant-superabsorption cycle, although the superabsorption occurs slightly less intensely than the superradiance ($10^{14} g^2$). Moreover, the delay time for superabsorption is slightly greater than that for the superradiance, the latter being around $\tau_D \approx 1.55 \times 10^{-3} g^{-1}$.

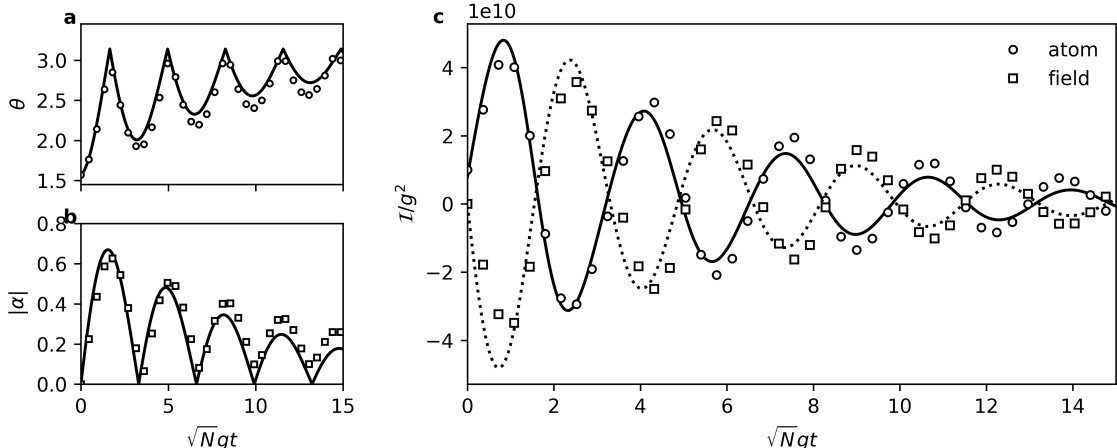

Figure 4: The same as in Fig. 2 for the underdamped regime: $N = 10^4$ and $\epsilon \approx 10$. Instead of $\theta_0 = 2/N$ as in Fig. 2, we have now considered $\theta_0 = \pi/2$.

In Fig. 4($a$, $b$, and $c$), we again plot the same functions as in Fig. 2, considering the underdamped regime for $N = 10^4$ such that $\epsilon \approx 10$. We again consider all other parameters equal to those in Fig. 2, except for $\theta_0 = \pi/2$ due to the linearization procedure. Now, we observe around 4 superradiant-superabsorption cycles, with intensities starting at around $10^{10}g^2$, as the strong coherence parameter leads to a slow damping of the initial atomic excitation. The number of superradiance-superabsorption cycles can be controlled by Stark shifting the sample out of resonance with the field. From Ref. [21] it follows that the time interval for a superradiant-superabsorption cycle is around two times the characteristic emission time $2/\sqrt{N}g$, which is in excellent agreement with Fig. 4($c$).

From Fig. 4($c$) it follows that the time required for 4 superradiance-superabsorption cycles is around $10/\sqrt{N}g$, resulting in the values $10^{-6}s$ and $10^{-10}s$, for the microwave and optical cavity-QED regimes, respectively. In the microwave regime the decay time of a high-finesse cavity is around a thousand times greater than $10^{-6}s$, while in the optical regime it is around 10 times greater than $10^{-10}s$, making it possible to carry out the experiment in both regimes, with advantage for microwave cavities.

We finally address the magnitude of the sample-field momentum transfer, assuming that the atomic spatial distribution is a narrow Gaussian centered around the node, $\Theta(x) = \exp\left[-x^2/2\sigma^2\right]/\sqrt{2\pi}\sigma$, of small enough width $\sigma$, such that $k\sigma \ll 1$. Considering, for the damped regime, the spatial distribution of the atomic sample of width $\sigma \approx 0.2/k$ and the parameters used in Fig. 3, we obtain a momentum transfer for the deflected atoms in states $|\pm, t\rangle$ around that of a cavity-field photon: $\Delta p \approx \pm k$. This magnitude is considerably smaller than the momentum uncertainty around $1/\sigma \approx 5k$. However the momentum transfer is significantly increased in the underdamped regime where a numerical account for the momentum distributions in Eq. (26) is shown in Fig. 5 against the scaled interaction time $\sqrt{N}gt$. The distributions $|\mathcal{F}_+|^2$ and $|\mathcal{F}_-|^2$ are represented by the green and red curves, respectively. We have considered the same parameter as in Fig. 4, with $\sigma \approx 0.2/k$, to observe that the momentum for $t = 6/\sqrt{N}g$ is around $50k$, far greater than the momentum uncertainty. As stated above, this momentum transfer is evidently greater the longer the sample-field interaction time.

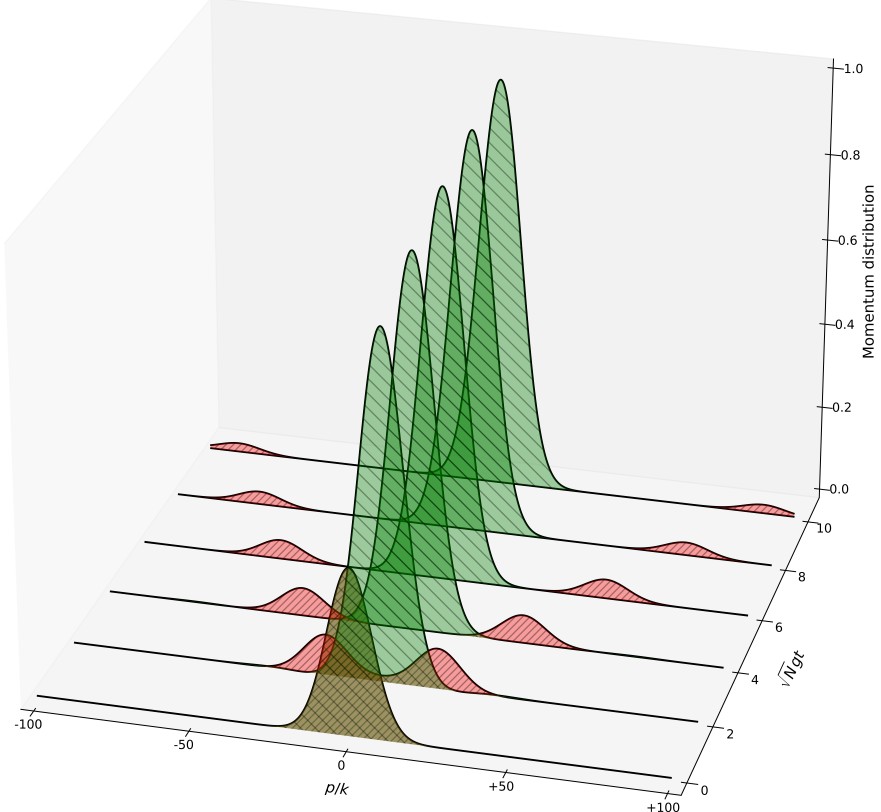

Figure 5: Plot of the momentum distribution functions $|\mathcal{F}_+|^2$ and $|\mathcal{F}_-|^2$, against $\sqrt{N}gt$, considering the same parameter as in Fig. 3, with $\sigma \approx 0.2/k$. $|\mathcal{F}_+|^2$ and $|\mathcal{F}_-|^2$ are represented by the green and red curves, respectively, in accordance with the colors in Fig. 1(d).

## 7 Conclusions

The use of the interplay between superradiance and superabsorption, advanced in Ref. [21], proves to be a suitable tool to achieve a coherent deflection of an atomic sample and consequently to achieve a long-sought goal: the preparation of momentum (or positional) mesoscopic superpositions. We stress that superradiant Rayleigh scattering from a Bose-Einstein condensate has been used to produce superpositions of (stationary) momentum states of recoiled atoms [32–34]. Such superpositions, created by the density modulation of the condensate and consequently the Bragg scattering regime, are different in nature from that in Eq. (27), where the momentum increases with time as observed in Fig. 5.

It is worth nothing that, in Ref. [21], a master equation was derived from which the Hamiltonian $H(t)$ of Eq. (2) comprised the von Neumann term. Since we are dealing with a short-time problem involving superradiance and superabsorption, we simply chose to disregard the terms of the irreversible evolution of the system associated with dissipative-diffusive effects. Taking these terms into account would make the analysis of the momentum transfer between the atomic sample and the field much more complex, without however adding significant gains. In addition to dissipation and diffusion (for finite temperatures), there is another decoherence effect in our proposal resulting from the dispersion in the atomic positions after the trap is turned off. This dispersion affects the approximation $kx \ll 1$, as time progresses, and consequently the coherence of the sample deflection.

The present proposal poses a challenge to the experimental physics of radiation-matter interaction, seeking to extend the remarkable advances achieved in the last 4 decades [17] to the domain of many-body physics. This has, in fact, already begun with the coupling of a Bose-Einstein condensates with a cavity field to achieve the Dicke quantum phase transition [35] and to enhanced superradiant Rayleigh scattering [36]. In particular, we observe that the present development, together with Ref. [21], can be used for the proposition of a more efficient quantum lithography protocol based on the deflection of atomic samples instead of individual atoms as in Ref. [10]. It can also be used for the construction of positional mesoscopic atomic entanglements, and for the implementation of quantum processing with mesoscopic ensembles, a goal that has been pursued since the early 2000s [37].

# Acknowledgments

**Funding information** The authors acknowledge financial support from CNPQ and CAPES, Brazilian agencies.

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
