# Peer review of "Coherent deflection of atomic samples and positional mesoscopic superpositions"

_SciPost Physics, doi:SciPost Phys. 18, 169 (2025)_

## Round 1 · Referee Report · Anonymous (Referee 1) · 2025-1-9

Strengths

1 - The manuscript by da Silva et al. presents a new protocol to achieve coherent deflection of an atomic sample coupled with the optical mode of a high finesse cavity, due to coherent momentum transfer. 2- the protocol can find application in the realization of mesoscopic atomic superpositions

Weaknesses

1- A simple, conceptual design of the system being studied is missing, only references to other papers are given.

2-no numerical example based on a realizable atomic system is given

Report

the paper is interesting and meets the criteria of the journal, before publication, I'd invite the authors to major changes as listed above.

Requested changes

While the paper is interesting and meets the criteria of the journal, before publication, I'd invite the authors to do:

1- spelling cross-check of the manuscript

Here are a few examples of misprints that I've found: l. 36 "antecipated" l.71 "rulled out" l.125 "senoidal functions" l. 128 "procedure" ...

2- cross-check definitions of measurable quantities introduced in the manuscript, as, for example in: - eq. 1 \omega and \omega_k not defined

For this, a schematic view of the experimental system under study might clarify.

3- Add practical, experimental numerical estimation of realizable systems to check in which conditions different effects/regimes (damped, underdamped, ...) might be experimentally observable.

Recommendation

Ask for major revision

  • validity: high
  • significance: good
  • originality: good
  • clarity: good
  • formatting: good
  • grammar: reasonable

Author:  Luís da Silva  on 2025-01-24  [id 5146]

(in reply to Report 1 on 2025-01-09)

Please, find in the File Attachment below our answers to the referee report #1.

Attachment:

Carta-Referee-SciPost-Physics.pdf

---

## Round 1 · Referee Report · Anonymous (Referee 2) · 2025-2-4

Report

The authors discuss the deflection of an atomic sample due to the momentum transfer from trapped atoms to a cavity field. The discussed protocol relies of the interplay between well known superradiance and superabsorption effects.

I checked the main calculations outlined along the text, and they looked me sound overall (I still have some questions, see below). However, I find the presentation style unsatisfactory, especially concerning the resulting clarity to read. In particular, I find unpleasant that the derivations of the main equations, although not so involved, are put together with the discussion of the results stemming from them. This fact makes the presentation quite tiring to follow, also because the relative weight devoted for the results turns out with almost a minor importance (that cannot be, clearly). The same problem holds for the description of preparation of the set-up, in my opinion. I also find inconvenient to locate all the pictures after the end of the text; perhaps this is intended just a feature of the preprint. However, this choice does not simplify the reading process.

For these reasons, I cannot recommend the manuscript for publication in the present form. I am open to reconsider it after an extended reconstruction of the text and after that my more specific criticisms (see below) have been properly addressed.

Best regards,

the referee.

———————————————

Specific questions:

  • how the author characterize the atomic decay factor ?

  • can the author address better (and separately) the (main steps of the) derivation of the MF equations ?

  • in the latter derivation, the \omega_k of the bath enter in the values of the averages on that ?

  • the expression for \epsilon after eq. 12 can be simplified in N, I guess.

  • the authors could recall briefly the basis general features of the 3 discussed regimes ?

  • in the same three regimes, can the author identify optimal numbers for the superradiance-superabsorption cycles ? How do these reflect on the allowed time-scales for the experiment ?

  • as far as I understand, the discussion is performed in the limit T \to 0. Can the authors include temperature effects (also from optical heating, perhaps) ? I have in mind also possible decoherence effects related to T. Other decoherence effects are expected ?

  • since, as correctly admitted by the authors, “the present proposal poses a challenge to the experimental physics of radiation-matter interaction”, can the authors themselves comment further on the required experimental strategies and developments ?

Recommendation

Ask for major revision

  • validity: good
  • significance: good
  • originality: good
  • clarity: ok
  • formatting: reasonable
  • grammar: excellent

Author:  Luís da Silva  on 2025-02-11  [id 5208]

(in reply to Report 2 on 2025-02-04)

Please, find in the File Attachment below our answers to the referee report #2.

Attachment:

Carta-Referee-2-SciPost-Physics.pdf

---

## Round 2 · Referee Report · Anonymous (Referee 2) · 2025-3-25

Report

The authors replied satisfactorily to my comments and criticisms. The manuscript results importantly improved. I recommend it for publication.

Recommendation

Publish (easily meets expectations and criteria for this Journal; among top 50%)

---

## Round 2 · Referee Report · Anonymous (Referee 1) · 2025-3-25

Report

The authors made substantial changes to the previous version, answering the main questions the referees raised. The manuscript can be published as is.

Recommendation

Publish (easily meets expectations and criteria for this Journal; among top 50%)

---

## Editorial Decision

published